# The importance of patient engagement in the multimodal treatment of MASLD

Joost Boeckmans [1,2], Hannes Hagström [1,3], Donna R. Cryer [4] & Jörn M. Schattenberg [5,6,7] ✉

Metabolic dysfunction-associated steatotic liver disease (MASLD) is often regarded in society as a disease caused by personal lifestyle and dietary choices. Healthcare providers who have empathy and are able to explain the disease trajectory can better engage with people with MASLD and actively work with them to improve their metabolic health on a sustainable basis. Non-invasive tests can assist in this process, but healthcare providers must ensure they explain their advantages and limitations. Discussing and setting lifestyle goals are priorities before initiating specific pharmacological treatment, since living a healthy lifestyle will remain the backbone of the multimodal management of MASLD. In this review, we discuss challenges and opportunities to actively engage with people living with MASLD in a multimodal treatment framework as a healthcare provider.

Non-communicable diseases (NCDs) account for 74% of global mortality, resulting in 41 million deaths annually. Cardiovascular diseases, cancers, respiratory diseases, and diabetes mellitus have been traditionally regarded as the 'big 4' NCDs since these conditions are responsible for the highest numbers of deaths[1].

Metabolic dysfunction-associated steatotic liver disease (MASLD) (formerly known as non-alcoholic fatty liver disease (NAFLD)) affects approximately one-third of the adult population[2] and is at the interface between multiple NCDs, including type 2 diabetes mellitus (T2DM), obesity, and cardiovascular disease[3]. A nationwide cohort study including 10,568 biopsy-confirmed patients with MASLD showed that patients with MASLD had an increased overall mortality rate compared to controls (28.6 versus 16.9/1000 person-years) caused by extra-hepatic cancers, cirrhosis, cardiovascular disease, and hepatocellular carcinoma (HCC)[4] with an average life-expectancy that is 2.8 years shorter[5]. As a result, MASLD also poses challenges to financial health care systems[6]. In addition, 9% of patients with MASLD experience stigmatization because of their liver condition, resulting in impairment of health-related quality of life[7] and potentially also healthcare avoidance[8]. Therefore, MASLD should feature prominently on the public health agenda[9,10].

MASLD encompasses a spectrum of disease stages ranging from isolated liver steatosis to progressive metabolic dysfunction-associated steatohepatitis (MASH), fibrosis, cirrhosis, and HCC, with only a minority of patients with non-cirrhotic MASLD experiencing more severe liver-related outcomes[11]. MASLD progression is driven by environmental and genetic factors and is part of the metabolic syndrome in which obesity and T2DM

are prevalent and important disease modifiers[12,13]. Societal factors contribute to the increasing prevalence of MASLD by promoting the development of these metabolic disorders through a sedentary lifestyle and encouraging ultra-processed food consumption (Fig. 1)[2,14]. Policy makers and the food industry hence have important roles in controlling MASLD in the population, although these are often affected by financial incentives. The most readily modifiable factors, depending on the stage of MASLD and comorbidities, involve personal behavior, including having a healthy diet and regular physical activity, and avoiding tobacco and alcohol use[15–17]. Since MASLD is a chronic and silent liver disease, there is limited knowledge about it in the general population which complicates effective patient communication and engagement[18]. In addition, patients with MASLD have limited readiness to adopt lifestyle changes, especially regarding exercise[19], while liver disease also often receives less attention compared to many other diseases[20]. This practical guide for the hepatologist and allied healthcare workers aims to provide concrete tips to encourage patients with non-cirrhotic MASLD to adopt lifestyle changes and improve their liver condition and overall metabolic health. Now the first drug to treat MASH, Resmetirom, has received approval by the Food and Drug Administration (FDA)[21,22], we also emphasize strategies to promote adoption of a healthy lifestyle on a sustainable basis whilst using MASH-specific drugs.

## Navigating in a relationship of trust
### Empathetic listening and building mutual trust
Because obesity might be the only visible indicator of MASLD, it can be challenging to present a strong message to patients, even more to those who

¹Department of Medicine, Huddinge, Karolinska Institutet, Stockholm, Sweden. ²In Vitro Liver Disease Modelling Team, Department of In Vitro Toxicology and Dermato-Cosmetology, Faculty of Medicine and Pharmacy, Vrije Universiteit Brussel, Brussels, Belgium. ³Division of Hepatology, Department of Upper GI, Karolinska University Hospital, Stockholm, Sweden. ⁴Global Liver Institute, Washington, DC, USA. ⁵Department of Medicine II, University Medical Center Homburg, Homburg and Saarland University, Saarbrücken, Germany. ⁶PharmaScienceHub (PSH) Saarland University, Saarbrücken, Germany. ⁷Centrum für geschlechtsspezifische Biologie und Medizin (CGBM), Saarland University, Saarbrücken, Germany. ✉e-mail: joern.schattenberg@uks.eu

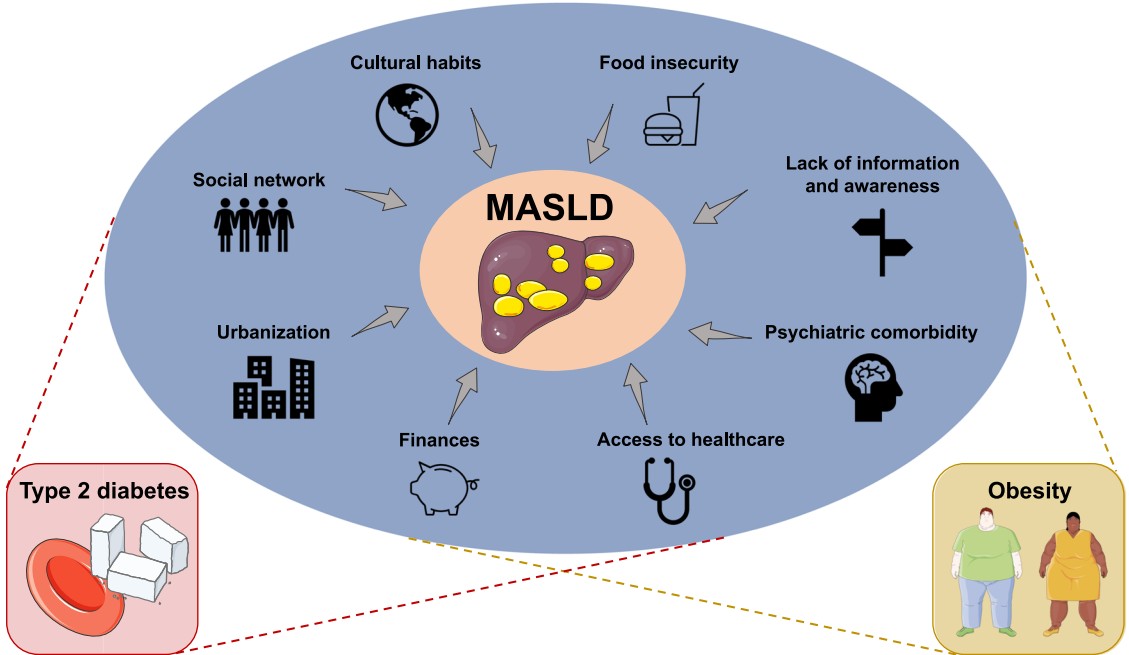

**Fig. 1 | Societal factors contributing to the development of MASLD.** Multiple non-biological factors can modify behavior and lifestyle choices that lay at the basis of MASLD development. Type 2 diabetes mellitus and obesity typically share these risk factors and are important disease modifiers in MASLD. MASLD metabolic dysfunction-associated steatotic liver disease.

are lean[23]. However, creating a safe environment and providing statements such as 'we will make a plan together to improve your liver condition, which will put you on track for a healthy life' can be motivational and give a sense of a provider-patient partnership striving for a better quality of life and reduced risk of both liver- and non-liver-related outcomes (Fig. 2). People with MASLD should therefore be seen as persons with lived experience rather than patients waiting for treatment. Listening to each story requires in depth attention and empathy but can yield mutual advantages through advancing insights into personal behavior. Patients can experience a feeling of importance and dignity, and this can be used as a solid foundation for making plans together using shared decision making[24,25]. Family or friends accompanying the patient can be involved in this process to assist in a supportive network on a daily basis[26].

### Establishing mutual strategies and identifying personal factors that hinder change to engage the patient to collaborate with other experts

Once a feeling of mutual trust and common goals is accomplished, personal factors that hinder adoption of lifestyle changes should be identified, for example through motivational interviewing[27,28]. Ideally, this process results in collaboration with other experts such as dietitians, cardiologists, endocrinologists, pharmacists, physiotherapists, nurses, psychologists, and/or social workers, as needed, leading to multimodal treatment of MASLD and associated diseases[29]. From a practical point of view, the central persons coordinating these partnerships would be the hepatologist and nurse in cases of more advanced disease, and the primary care provider for patients with early stage MASLD. Monthly or quarterly multidisciplinary meetings could be organized to evaluate the goals and needs of the individual patients.

### Constructive approaches to address failures to adopt lifestyle changes

During the process of changing lifestyle, failure should be seen as an opportunity to identify points to improve and as a step in the right direction as this is more motivating than viewing it as personal malfunctioning[30,31]. Since nearly all patients will encounter relapses when changing their lifestyle and moments of uncertainty, it can be explained that these are non-linear achievements of the goal and integral elements of their process, and that

changing lifestyle requires learning and reflecting on what can be improved (Fig. 3). Numbers related to lifestyle and weight loss that appeal to the imagination are available for MASLD and sharing these can assist in this process. Patients able to lose 10% or more of their body weight through lifestyle modifications show reductions in the NAFLD activity score after 1 year, while 90% show resolution of MASH and 45% even have regression of hepatic fibrosis[32]. It must however be noted that among people living with overweight, only approximately 20% can achieve and maintain a 10% intentional weight loss for at least 1 year[33,34]. Maintaining weight loss is therefore perhaps the most challenging part of adopting lifestyle changes in patients with MASLD and some of them might never return to a non-overweight/obese state. For these individuals, it is essential to affirm that any weight loss still adds significant benefits to their health, and that gradual weight reductions can sometimes be easier to adhere to than drastic changes in lifestyle[35,36]. In addition, discussing the differences between body weight and body composition[37] is crucial for patients who are improving their health through physical exercise without achieving weight loss.

### Considering the role of society

Although multimodal treatment of MASLD is a good approach[29], one needs to be cautious with patients who feel stigmatized and guilty about their metabolic condition and are potentially frustrated by a virtual environment portraying a beauty ideal, elite athleticism, sexuality, and misguided appearance of health[7,20,38]. Today, a sedentary lifestyle is supported by innovation, industrialization, and urbanization, and the impact of this on someone's daily life can be enough to develop diseases related to metabolic dysfunction (Fig. 1). To that end, a Sustainable Development Goal score has been developed as an advocacy tool for NCDs, including MASLD. The Sustainable Development Goal score for MASLD provides an estimate of the country-level preparedness to manage MASLD from a societal perspective, which can facilitate multisectoral collaborations. Indicators for sustainable development regarding MASLD are child wasting, child overweight, NCD mortality, a universal health coverage service coverage index, health worker density, education attainment, and an urban green space indicator, which is important for physical and mental health. Each of these factors can negatively or positively influence the development of MASLD, highlighting the role of society, education, and upbringing. For example,

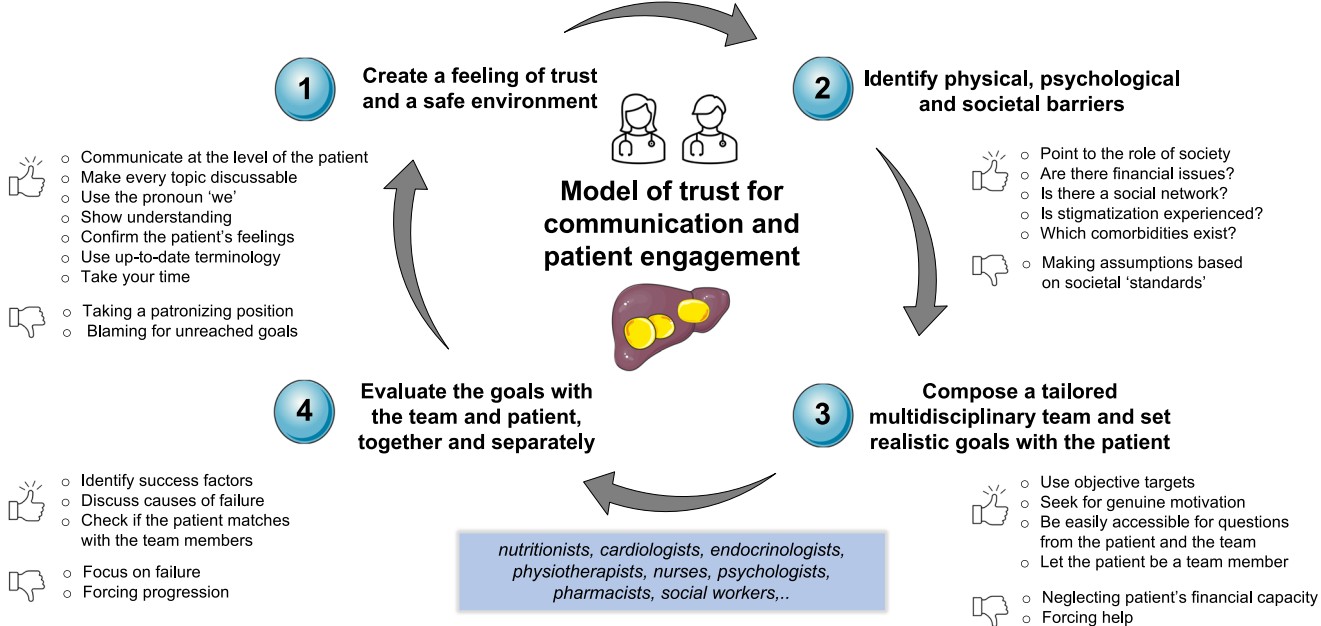

**Fig. 2 | Model of trust for communication and patient engagement in MASLD.**
Patients suffering from MASLD may either experience difficulties in communicating about their liver disease because of societal stigmatization or have a lack of motivation to actively work on their metabolic health. Creating a safe environment in which up-to-date terminology is used with attention for psychological and societal inequalities seems key for effective communication in MASLD to attain sustained lifestyle changes. A multidisciplinary team with continuous evaluation should therefore be composed based on individual needs. MASLD metabolic dysfunction-associated steatotic liver disease.

mortality due to T2DM and cardiovascular disease might not have been a health priority in low- and middle-income countries that were instead focused on communicable diseases. Such factors are captured by the Sustainable Development Goal score, which can subsequently be used to inform policy makers to take action at the population level in an objective manner[38].

A holistic approach to communicating the need for lifestyle changes can allow patients to realize that there are forces working against them as they aim for better health outcomes. Highlighting the societal perspective can also make a patient more aware of lifestyle- and food-related signals in daily life. Digital therapeutics including internet applications specifically designed for patients with MASLD can be proposed to assist in this process. Such applications are tailored to the individual's needs and assess motivation to change and consciousness of their disease. Further, they provide education using interactive slides and can guide physical exercise[39,40]. However, social support is the greatest facilitator for making lifestyle changes in patients with MASLD[41]. From a social and societal perspective, another approach to promote lifestyle changes lays in social prescribing. Patients can be 'prescribed' non-medical activities to promote their metabolic health utilizing local initiatives based on their personal interest and motivation. These can consist of joining a local sports club, gardening group, or cooking club focused on healthy eating habits. This way, patients can meet other facets of society, join new social networks and discover health-promoting hobbies that match their personality[42,43]. In addition, it offers an elegant way to circumvent the general sports advice while still empowering physical activity and healthy habits. Patients with MASLD can reside in a microenvironment that promotes the development of MASLD, so sharing these interventions with family members and cohabitants can yield metabolic benefits beyond the individual patient, while also creating a supportive atmosphere[44].

**Considering psychiatric comorbidity**
A pitfall in MASLD patient engagement lays in the fact that depression is a highly prevalent condition among patients with MASLD (prevalence of 18.21% (95% CI 11.12;28.38) for MASLD and 40.68% (95% CI 25.11;58.37) for MASH)[45,46], which can impede motivation to work on lifestyle-related factors. Since depression is often treated in primary care, it is important to also involve general practitioners in the

interdisciplinary team. Nonetheless, it remains vital to identify details that are suggestive for a clinically relevant or subthreshold depression since it would require involving specific care[47]. Furthermore, several antidepressants and other drugs used for psychiatric diseases are appetite-promoting and induce weight gain[48], which can be problematic. It is the shared responsibility of the prescribing physician and pharmacist to select the least weight-inducing agent for a specific patient, and to inform the patient of this potential side-effect, in particular when the patient is overweight or obese. A potential future avenue consists in the additional prescription of a glucagon-like peptide-1 (GLP-1) receptor agonist to pro-actively address expected weight gain induced by psychiatric drugs[49,50].

Apart from depression, there is also a relationship between MASLD and the development of anxiety disorders, in particular in women (hazard ratio for women 1.29 with 95%CI 1.13;1.48—hazard ratio for men 1.15 with 95%CI 0.99;1.34)[51], which can further complicate patient communication and engagement. For these patients, emotional support and creating a safe space for sharing experiences and fears are even more important. Interaction with, and obtaining accurate details from, these patients can be achieved by showing curiosity and requesting the patient to correct you if something feels not right or sounds unclear[52].

Once compensated advanced chronic liver disease/cirrhosis has developed, one should be attentive for signs of hepatic encephalopathy, a neuropsychiatric disorder characterized by confusion, cognitive impairment, poor concentration, and changes in personality and behavior[53]. These symptoms can be relatively easily linked to advanced liver disease together with other signs and symptoms, while the true difficulty lies in the recognition of minimal hepatic encephalopathy (MHE), in which only subtle symptoms occur related to vigilance and integrative function[54]. Although some reports exist on cognitive impairment caused by MASLD[55], MHE should not be ignored as it relates to impaired quality of life, frequent road traffic accidents and a poor prognosis. In addition, MHE is present in 35% of patients with cirrhosis, making it a prevalent condition[53]. Testing for MHE can be undertaken via checks such as the psychometric hepatic encephalopathy score (PHES) if communication is impaired by possible symptoms[56]. Since the PHES is a time-consuming test, a simplified animal

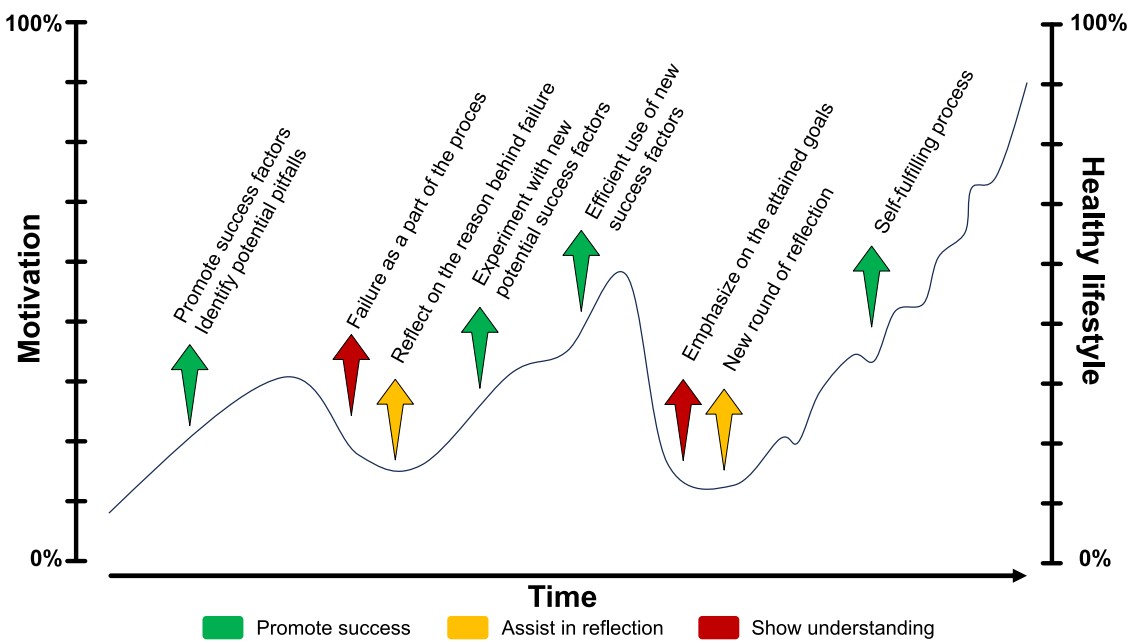

**Fig. 3 | Process of adopting lifestyle changes.** Helping patients identify and leverage strengths and reflect on the causes of failure can result in sustainable adaptive behavior towards metabolic health.

naming test can be used instead in daily clinical practice, although it is less specific[57].

## Engagement by information

### Importance of being aware of changing terminology

Three different acronyms have been used in recent years to describe in essence the same disease entity: NAFLD, metabolic dysfunction-associated fatty liver disease (MAFLD), and MASLD[58]. This has not only led to confusion in the scientific literature, but also heterogeneity in information sources available to the public. A key element to maintaining credibility in the patient-physician relationship when communicating with patients is consistency in what is being told to them. Since June 2023[12], professional societies including the American Association for the Study of Liver Diseases (AASLD), the European Association for the Study of the Liver (EASL) and the Latin American Association for the Study of the Liver (ALEH) have undertaken a consensus process among stakeholders and advised that MASLD will be the term to be used in future healthcare activities. This change in terminology should be explained to people with MASLD. For example, newly diagnosed patients with MASLD are likely to encounter patient documentation in which the NAFLD (or MAFLD) nomenclature is used. Vice versa, patients who have previously received a diagnosis of 'NAFLD' should be educated on the new MASLD terminology so they can find up to date information about their disease.

The change in MASLD nomenclature constitutes an opportunity for physicians to motivate and inform their patients. In this nomenclature, the potentially stigmatizing terms 'fatty' and 'alcoholic' were removed, and the role of metabolic dysregulation was highlighted[12]. Further, it is in line with the preferred communication by the subset of patients with MASLD and obesity, who opt for terms such as 'weight' rather than 'fat' or 'obese'[23,59].

### Education on prognosis

The unawareness in the population regarding liver disease in general and more specifically MASLD also results in a lack of basic knowledge about the disease and more importantly, its long-term consequences. Sharing prognostic details on MASLD can engage patients and encourage the introduction and maintenance of behavioral changes[9,60]. The most easily explained disease perspective is perhaps the suggested '20% rule' for progression in F3/F4 MASH, stating that 20% of patients with MASH and bridging fibrosis develop cirrhosis in 2 years, while 20% of patients with cirrhosis develop hepatic decompensation in 2 years[61]. Yet, being aware you have liver fibrosis can potentially already promote a healthier lifestyle, since unawareness about liver fibrosis stage by people with MASLD/MASH has been shown to be associated with poor adherence to lifestyle changes[62]. Nonetheless, most patients have isolated liver steatosis and might never develop MASH and more advanced liver disease and having a 'fatty' or 'steatotic' liver is often considered as being a condition without severe hepatic consequences. To these patients, it can be explained that MASLD is an important contributor to T2DM and cardiovascular disease with potential mortality[63–66]. In addition, HCC can also develop in patients with MASLD in the absence of cirrhosis[67,68], which can be a strong call to action. In light of these complications, it is of importance to link this chance of progression to the ability to change the disease course by decisive action and life-style choices. Therefore, education on the natural history of MASLD and the reversibility of liver steatosis, MASH, and fibrosis through adopting lifestyle changes are key aspects to promoting intrinsic motivation and preventing the terminal complications of MASLD[32,69]. Educational material from medical associations (such as EASL[70]) and patient organizations can assist in this process. In line with this, an initiative to inform patients with MASLD is the 'Global Fatty Liver Day', which is a public education campaign supported by liver patient organizations and multiple medical societies[71].

Apart from these liver-related perspectives, pregnancy-related and inter-generational factors can also be used to promote behavioral changes for specific individuals. MASLD is independently associated with hypertensive complications (pre-eclampsia, eclampsia, and/or HELLP syndrome (Hemolysis, Elevated Liver enzymes and Low Platelets)) (odds ratio 3.13 with 95%CI 2.61;3.75), postpartum hemorrhage (odds ratio 1.67 with 95%CI 1.28;2.16), and preterm birth (odds ratio 1.60 with 95%CI 1.27;2.02), invigorating the need for pre-conception counseling[72]. In addition, maternal obesity increases the risk of MASLD in the offspring (odds ratio 3.26 with 95%CI 1.72;6.19)[73], which is an additional argument for adopting lifestyle changes in potential future mothers.

### Role of non-invasive tests

As patients generally prefer to have direct access to their medical results[74], even when these are normal[75], non-invasive test (NIT) results

that screen for MASLD phenotypes including the fibrosis-4 (FIB-4) score and vibration-controlled transient elastography (VCTE)-based scores, using the controlled attenuation parameter (CAP) and liver stiffness measurement (LSM), can be convenient tools to inform patients about their liver status and initiate discussion (Table 1)[29]. Nonetheless, as highlighted in Table 1, these test results can be complex to interpret[76], which can lead to unfounded worries in patients[74]. Moreover, small changes in these NITs, including CAP and LSM, do not reflect histological or clinically relevant improvements[77,78], indicating the importance of targeting clinically relevant changes. For example, patients with a high Agile3+ score at baseline should attain a decrease in their score of more than 20% to have a considerable reduction in the risk of liver-related events[79]. In addition, the inaccuracy of CAP in differentiating higher steatosis grades can be discouraging in patients starting from steatosis grade 3[80]. More accurate NITs to quantify hepatic steatosis and fibrosis over time include magnetic resonance imaging-proton density fat fraction (MRI-PDFF)[81] and magnetic resonance elastography (MRE)[82], respectively. These could be valuable alternatives but their cost-effectiveness for this purpose is unclear[83]. Therefore, exact results from currently used NITs should be used with caution in patient communication, although highlighting improvements in these tests over time might be further motivational. In addition, their use in patient communication promotes patient participation which can positively influence patient-reported[84], and clinical outcomes, including liver- and cardiovascular events[85]. A potential alternative to sharing NIT data is showing pictures of livers[86].

Although NITs specifically designed for liver disease can encourage patient participation and provide information, one should keep in mind that changes in these parameters can lag behind more commonly used laboratory measurements of metabolic health. Therefore, earlier changes in for example blood pressure, cholesterol, triglycerides, and HbA1c can be used for achieving shorter-term goals while also working towards metabolic health in general[85,87,88].

## Empathizing with different communities
### Consideration of socio-economic status and education
Socio-economic status is a risk factor for NCDs, mediated by a low-quality diet, living a sedentary lifestyle and a lack of higher education[89], and this also applies to MASLD[89,90]. A low socio-economic status goes hand in hand with food insecurity, which is a risk factor for MASLD. In this regard, 'food deserts', areas with sparse options to acquire nutritious food, and 'food swamps', areas with a high concentration of fast food- and junk food-selling restaurants, create an obesogenic climate that promotes the development and worsening of MASLD[91]. In line with this, low socio-economic status may also limit access to physical activity options and exercise routines because of costly memberships or time constraints[92].

Access to healthcare is a prerequisite for patients with a lower socioeconomic status to adopt lifestyle changes, which should not only be available, affordable, and acceptable, but also sustainable[93]. A silent liver disease does not inspire change in many people[94], especially when other issues in life, among which securing food and financial stability, tend to be a higher priority. Dietitians and social workers can play a key role in aiding the transition to eating healthy food, although this may be adopted at the expense of financial burden. Referral to these professionals can therefore be communicated to a patient by emphasizing the goal of a tailor-made nutritional plan, taking into account budgetary constraints and practical feasibility, including transportation. Informing

**Table 1 | Considerations when sharing non-invasive test results with patients for hepatic steatosis and fibrosis in MASLD**

| Hepatic steatosis | | | |
|---|---|---|---|
| **Test** | **Test type** | **Consideration** | **Ref.** |
| Fatty liver index | Blood-based | >Differentiation of higher steatosis grades is inaccurate<br>>Indeterminate zone (30;60) difficult to interpret<br>>Not liver-specific | 144–146 |
| Ultrasound | Imaging-based | >High inter- and intra- observer variability, limited reproducibility<br>>Low accuracy for grading steatosis | 147,148 |
| VCTE - CAP | Imaging-based | >Differentiation of higher steatosis grades is inaccurate<br>>Moderate accuracy in patients with obesity | 80,143,149,150 |
| **Hepatic fibrosis** | | | |
| **Test** | | **Consideration** | **Ref.** |
| Fibrosis-4 score | Blood-based | >Inaccurate in people younger than 35 years<br>>Unspecific in patients older than 65 years<br>>Can be false positive when another cause of thrombocytopenia is present, such as HIV-infection<br>>Better at excluding than including advanced fibrosis<br>>Indeterminate zone (1.3;2.67) difficult to interpret<br>>Results can be impacted by liver congestion<br>>Not liver-specific | 88,151–153 |
| NAFLD fibrosis score | Blood-based | >Inaccurate in people younger than 35 years<br>>Unspecific in patients older than 65 years<br>>Better at excluding than including advanced fibrosis<br>>Indeterminate zone (-1.455;0.676) difficult to interpret<br>>Not liver-specific | 151,154 |
| Enhanced liver fibrosis score | Blood-based | >High sensitivity but limited specificity for excluding significant/advanced fibrosis at low cutoffs<br>>High negative predictive value<br>>Influenced by age, inflammation, and matrix turnover | 155,156 |
| VCTE - LSM | Imaging-based | >Results can be impacted by liver congestion, ascites, active hepatitis, food intake, biliary obstruction, and amyloidosis<br>>Results are less reliable in people with severe obesity<br>>Better at ruling out than ruling in cirrhosis<br>>Results obtained with the XL-probe are often lower than those obtained with the M-probe<br>>CAP values higher than 300 dB can overestimate LSM in case of low fibrosis stage<br>>Differentiating lower fibrosis stages is inaccurate | 149,157–159 |

*CAP* controlled attenuation parameter, *LSM* liver stiffness measurement, *MASLD* metabolic dysfunction-associated steatotic liver disease, *VCTE* vibration-controlled transient elastography.

a patient that financial issues have been discussed beforehand with the health professional, can give the patient reassurance to effectively make use of these consultations and advice.

## Consideration of ethnicity and cultural background

Ethnical origin is often discussed in relation to genetic predisposition to MASLD, in particular the polymorphism in the patatin-like phospholipase domain-containing protein 3 (PNPLA3), rs738409[95], which is associated with a higher frequency of MASLD in Hispanics[96]. Ethnicity and cultural background can also have indirect effects on MASLD through lifestyle choices and dietary patterns[97,98] and so is a consideration when communicating with patients who have a different ethnical/cultural background than their physician. In this regard, assumptions made from the physician's own perspectives should be avoided to obtain an accurate view of someone's lifestyle and dietary pattern. Standardized questionnaires about food intake can assist in the process of identifying points of potential improvement[99,100], after which culturally tailored adaptations to diet can be made by suggesting foods that patients are familiar with that could allow sustainable behavioral changes[101–103]. Digital devices providing information about body composition and artificial intelligence-guided personalization of digital tools that factor in a person's own habits and culture can potentially aid in this process[104].

## Roles of alcohol consumption and smoking
### Reducing alcohol consumption
No amount of alcohol consumption improves health[105], but much heterogeneity exists in the conclusions of studies on the effects of alcohol consumption on MASLD, with some reporting that modest alcohol consumption protects against MASH[106] and some stating that any level of alcohol use should be avoided by patients with MASLD[107]. Nonetheless, there is consensus that alcohol consumption should be strongly discouraged in patients with MASLD who have F3-F4 fibrosis[108]. The recent introduction of the SLD nomenclature affirmed the added pathogenic value of alcohol in MASLD by introducing MetALD, in which patients have SLD originating from both metabolic dysregulation and alcohol consumption of 140 to 350 g/week for females and 210 to 420 g/week for males[12]. This new disease category not only allows better classification of patients, but its use can also motivate patients to limit their alcohol consumption. The first important steps in avoiding or limiting alcohol use consist of depersonalizing drinking habits and explaining potential negative liver- and non-liver related health outcomes. In a second step, barriers can be identified that hinder efforts to stop drinking[109]. These barriers often include social drinking and family habits, which are generally considered as relatively innocent drinking moments[110]. Social drinking can be reduced by sincerely expressing concerns and agreeing on a plan to reduce alcohol use[111]. For family habits, such as daily alcohol consumption during dinner, including other family members in the discussion can be beneficial[112].

### Reducing tobacco use
Smoking has been associated with MASLD in several studies (with an odds ratio for MASLD when smoking of 1.11 with 95%CI 1.03;1.20)[113–115]. Initially, patients can be told of the negative effects and risks of smoking on their metabolic and cardiovascular health[116], after which the potential impact on their partner or children can be discussed as well in certain cases. Although such a direct communicative strategy may lead to self-stigmatization, it has been reported that it can result in reductions in smoking[117]. For patients with T2DM and MASLD it can be further emphasized that the combination with tobacco use adds to the risk of hepatic fibrosis, while it is a potentially readily changeable factor when compared to T2DM (prevalent fibrosis odds ratio for cigarette smoking and T2DM interaction = 3.04 with 95%CI 1.62;5.76; odds ratio for T2DM alone = 2.28 with 95%CI 1.37;3.85)[118].

## Considering patient's expectations and preferences
Multidisciplinary healthcare teams have been increasingly proposed in recent years to address the complexity of MASLD and multiple comorbidities[29,41]. However it is important to aim for patient-centered communication and minimally disruptive care, so the expectations patients have and their preferred level of care should be considered[119,120]. Taking time to listen to these expectations in the very first consultation can save considerable undesired and therefore ineffective efforts and costs. In addition, patient satisfaction is generally driven by the feeling that their physician provides enough time to listen to understand their situation, which is essential to maintain the patient-physician relationship[121,122]. Credibility to patients can be strengthened by also communicating their expectations and preferences to the other healthcare professionals involved, so a feeling of immediate common commitment can be attained.

On the other hand, some patients might not have the motivation to set goals and work together towards metabolic health. If patients' intrinsic engagement is thought to be insufficient to achieve improvements in their liver condition, one should not be afraid to acknowledge the emotional and structural factors that may need additional support. This can enable hindering factors to be identified and made discussable, potentially resulting in improved motivation. Such conversations should be undertaken on a regular basis as personal engagement can change over time and different opportunities arise on different occasions. In line with expectational follow-up, regularly ascertaining satisfaction with the treatment plan and professionals involved will further augment the chances of success[123]. There exist, for example, diverse needs for behavioral support for successful lifestyle change which could range from peer support to coaching to structured psychology and psychiatry needs both at the inter-individual and intra-individual level over time[47,89]. For example, a meta-analysis investigating the effect of cognitive behavioral therapy on lifestyle changes found that it improves weight loss (effect size (BMI) −0.63 with 95%CI −1.17; −0.10) and weight maintenance (effect size (BMI) −0.55 with 95%CI −0.90; −0.20)[124], which are key to treating MASLD[32].

Furthermore, it remains important to underscore that MASLD is a slowly progressing disease and that there is space to take incremental small steps towards a metabolically healthy condition to avoid progression to cirrhosis[61], which can be reassuring.

## The impact of MASH-specific drugs
The lack of an effective pharmacological treatment for MASH[125] and the drastic nature of bariatric surgery[126] can engage patients to adopt lifestyle changes. Recently, Resmetirom, a thyroid hormone receptor-beta agonist, was approved by the FDA as the first drug to treat MASH[21,127]. The availability of this drug will enable liver-related goals to be more easily attained, and this needs to be considered to ensure an optimum patient-physician relationship. Nevertheless, lifestyle changes will remain essential to achieve holistic better metabolic health, including cardiovascular benefits, and avoid sarcopenia[128,129], and discussing these issues is a priority before initiating pharmacological treatment (Fig. 4). It can be helpful to also emphasize that Resmetirom was approved as an addition to diet and exercise and not as standalone treatment[130].

Improved health-related quality of life related to drug treatment, as observed in a 36-week phase 2 trial with Resmetirom[131], can be used as an opportunity to introduce new dietary and physical exercise habits during drug treatment, given that a better quality of life may come with enhanced motivation[132,133]. In that regard, GLP-1 receptor agonists that are indicated for the treatment of T2DM, also often induce considerable weight loss[134,135], which can eliminate obesity-related stigmatization and motivate patients to further work on their metabolic health through lifestyle modifications. In a 72-week randomized phase 2 trial, the GLP-1 receptor agonist Semaglutide induced significantly more MASH resolution (up to 59% in the highest dose (0.4 mg) group, compared to 17% in the placebo group, $p < 0.001$), but no statistically significant

**Fig. 4 | Importance of lifestyle changes in the era of MASH-specific drugs.** Adopting and continuously evaluating lifestyle changes is a priority when initiating MASH-specific drugs to attain metabolic health on a sustainable basis. Therefore, multimodal treatment of MASLD remains an important element even when MASH drugs are available. MASH metabolic dysfunction-associated steatohepatitis, MASLD metabolic dysfunction-associated steatotic liver disease.

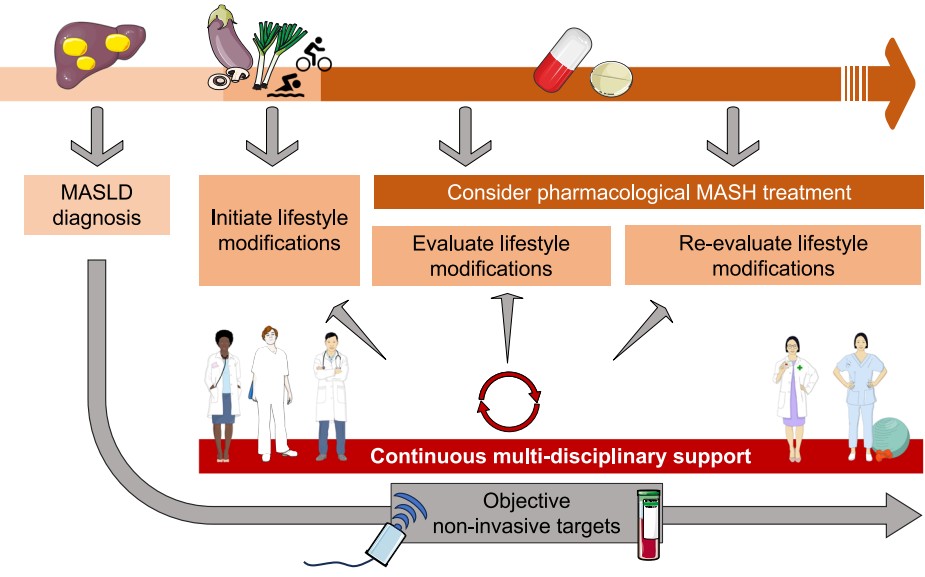

---

## Box 1: | Key elements in the multimodal treatment of MASLD. [MAFLD metabolic dysfunction-associated fatty liver disease, MASH metabolic dysfunction-associated steatohepatitis, MASLD metabolic dysfunction-associated steatotic liver disease, NAFLD non-alcoholic fatty liver disease]

- MASLD is a prototypical non-communicable disease and situated at the interface of obesity and type 2 diabetes mellitus.
- Empathy, authenticity, and education are key to patient communication in MASLD and are required to attain a sustainable patient-physician relationship.
- Consistency in patient information sources, including social media and self-obtained information, is important, which can be hindered by the multiple disease names and acronyms used in recent years, including NAFLD, MAFLD, and MASLD.

- Multi-disciplinary support should be tailored to the patient's specific needs and education, and its composition should be evaluated over time.
- Non-invasive tests can be used to monitor MASLD, but their use requires education on their specific advantages and limitations in assessing liver health.
- A healthy diet and physical exercise remain the basis of metabolic health in the era of MASH-specific drugs.
- Psychological benefits obtained by using MASH-specific drugs can potentially offer the opportunity to introduce new lifestyle modifications.

---

improvement in fibrosis stage, compared with placebo (43% in the 0.4 mg group, compared to 33% in the placebo group, $p = 0.48$). Nonetheless, GLP-1 receptor agonists (and dual/triple agonists with additional agonism of the glucagon receptor and/or glucose-dependent insulinotropic polypeptide receptor) might pave the way for future combined treatment of MASH and obesity[128,135,136] while also potentially having destigmatizing effects with resulting societal benefits. So far, GLP-1 receptor agonists are only regionally registered for the treatment of obesity and there exists much inequality in access to it due to requirements for access and lack of financial coverage[137,138].

### Conclusions and perspectives

MASLD is an increasingly prevalent health problem[139]. Lifestyle changes are the most readily available and best treatment currently available for MASLD, but these require a careful communicative strategy (Box 1). Mutual trust in the patient-physician relationship and involved healthcare providers in a multidisciplinary team are key to enable patients to reverse MASLD and prevent the progression to advanced disease. However, MASLD should not only be managed at the individual patient level, but also more comprehensively at the political level[9,10]. The EASL-Lancet commission[10], the 'Healthy Livers Healthy Lives' coalition[139], and 'Liver Health is Public

Health' initiative[140] have highlighted the need for better liver care and preventive strategies to be on the agenda of policy makers and provide examples of how to develop a public health strategy.

It is often highlighted that there exists unawareness regarding MASLD in the general population, even in patients who have obesity and T2DM[18]. Yet, this unawareness reaches beyond the general population as the relevance of MASLD and its impact on related comorbidities are also often poorly known in specialty disciplines outside hepatology[29]. This unawareness in medical care also hampers effective communication of a unified message to patients with MASLD. One of the most important health care professionals who need more knowledge about MASLD are general practitioners, since these physicians are responsible for initial diagnosis of MASLD and referral to specialty care. In recent years, much effort has been made to create efficient and practical referral pathways from primary care to specialty care, by, for example, using the FIB-4 score as a screening modality for advanced hepatic fibrosis in patients suffering from pre-T2DM or T2DM[141]. A concern when setting up referral pathways is that it may result in large numbers of patients needing specialty care, resulting in long waiting lists to access this care. Nonetheless, relying solely on primary care for patients with MASLD and initiating a holistic treatment plan involving a tailored

multidisciplinary team is not feasible due to time constraints and specialized aspects of follow-up. To enable a practical structure in secondary and tertiary care that allows communicating MASLD from a societal, psychological, and socio-economic perspective with patients, along with a resulting treatment plan, governmental reimbursement should be foreseen to finance the management of the team and contributions of healthcare professionals making part of it, including dietitians and psychologists[142].

As obesity and MASLD are potentially stigmatizing[7], the use of NITs can partly eliminate self-blaming and provide objective targets when adopting lifestyle modifications. However, there is no consensus on which NIT should be used to monitor disease and inform patients over time. Using the FIB-4 score would be a convenient way to allow general practitioners to evaluate disease over time since it only requires assessment of transaminases, platelet count, and age. However, the FIB-4 score remains a screening tool and fluctuating results in the indeterminate range (FIB-4 score 1.3 – 2.67) may be difficult to interpret and lead to unnecessary worry[77,78]. Given the substantial costs and limited availability of MRI-PDFF and MRE to assess hepatic steatosis and fibrosis, respectively[82,143], VCTE-based measurement of CAP and LSM seems to offer a better balance between accuracy, accessibility, and costs[80]. A VCTE-based follow-up could also be partly implemented in primary care to promote accessible liver-oriented healthcare, patient-centered participation, and consistency with specialty care.

The recent FDA approval of Resmetirom in the United States[21] has been of considerable interest to people with MASH/MASLD. As observed with the GLP-1 receptor agonist Semaglutide, which is registered in most countries for the treatment of T2DM and causes weight loss as an additional benefit[135], metabolic goals are more easily achieved by patients with concurrent obesity and T2DM. Yet, metabolic health requires a healthy diet and physical exercise, and metabolic targets should always be viewed in this perspective. As a result, one might question whether certain metabolic goals obtained through patient engagement should be achieved, or at least attempted, before specific drug treatment can be initiated. Adopting lifestyle modifications could be used as a justification for subsequent pharmacological treatment. Conversely, specific treatment for MASH could potentially break a cycle of unhealthy dietary patterns.

In conclusion, effective communication to engage patients with MASLD over the long term will only be achievable if there is open communication, genuine trust, and mutual appreciation[122] taking into account societal and mental issues as well as the biological causes of the disease[38]. With the availability of MASLD/MASH-specific drugs[21], it will remain of utmost importance to maintain this relationship to be able to stimulate a healthy diet and physical exercise to obtain sustainable metabolic health.

In the next 5 years, we speculate that patient engagement in MASLD will be enabled in many countries through dedicated multimodal treatment plans.

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

## Acknowledgements
Figures were made using Servier Medical Art (licensed under CC BY 4.0). J.B. receives funding from Onderzoeksraad Vrije Universiteit Brussel and Chair Mireille Aerens for the Development of Alternative Methods. J.B. and J.M.S. take part in the EASL mentorship programme.

## Author contributions
J.B.: conceptualization, visualization, methodology, interpretation, writing—original draft, writing—review and editing. H.H.: interpretation, input of critically important information, writing—review and editing. D.R.C.: interpretation, input of critically important information, writing—review and editing. J.M.S.: conceptualization, supervision, project administration, methodology, interpretation, writing - review and editing.

## Funding

## Competing interests
J.B. reports research funding from Colgate-Palmolive. H.H. reports research funding from AstraZeneca, EchoSens, Gilead, Intercept, MSD, Novo Nordisk and Pfizer. He has served as a consultant for AstraZeneca and Novo Nordisk, and has been or is part of hepatic events adjudication committees for Arrowhead, Boehringer Ingelheim, KOWA and GW Pharma. D.R.C. is an employee of the Global Liver Institute, which convenes the NASH Council, has received grants and sponsorships from several companies in the NASH therapeutic space, and is an advisor to PathAI and Chronwell. J.M.S. reports consulting for Alentis, Alexion, Altimmune, Astra Zeneca, 89Bio, Bionorica, Boehringer Ingelheim, Gilead Sciences, GSK, Ipsen, Inventiva Pharma, Madrigal Pharmaceuticals, Lilly, MSD, Northsea Therapeutics, Novartis, Novo Nordisk, Pfizer, Roche, Sanofi, and Siemens Healthineers. speaker honorarium from AbbVie, Academic Medical Education (AME), Boehringer Ingelheim, Echosens, Forum für Medizinische Fortbildung (FOMF), Gilead Sciences, MedicalTribune, MedPublico GmbH, MedScape, Novo Nordisk, Madrigal Pharmaceuticals, Stockholder options: AGED diagnostics, and Hepta Bio.
