## [Peer Review file · Communications Medicine]

The importance of patient engagement in the multimodal treatment of MASLD

Corresponding Author: Professor Jörn Schattenberg

Version 0:

Reviewer comments:

Reviewer #1

(Remarks to the Author)

To the editor,

It was a pleasure reviewing the manuscript entitled "Patient Engagement in the Multimodal Management of MASLD" by Boeckmans et al.

I think we can all agree that there is a wealth of ever-growing literature on healthy lifestyle modification strategies and pharmacotherapeutics to treat MASLD. Clinical trial data and real-world clinical experience also show that only some patients succeed in losing the recommended weight to achieve the desired endpoints of steatohepatitis resolution and hepatic fibrosis stabilization.

What seems to be missing, in my opinion, are papers outlining potential factors affecting patients' successes and failures in changing their lifestyles to improve metabolic abnormalities, including MASLD, and achieve optimal metabolic health. I believe that this incisive summary by Boeckmans, et al on patient engagement, in the MASLD population specifically, bridges this knowledge gap and provides quite practical and valuable guidance for the clinician taking care of MASLD and other metabolic conditions.

The following are my comments:

Part 2.1. Process of success and failure

I suggest that they divide this section into separate paragraphs outlining their points to organize the points better. For example, one paragraph is on emphatic listening and building mutual trust. Another on establishing mutual goals and plans and identifying personal factors that hinder change to engage the patient to collaborate with other experts. Another on constructive approaches to failure, providing meaningful feedback, and framing the importance of small successes.

Feedback on success can also be seen through changes in liver chemistries, liver stiffness and CAP score, lipids, and glycemic control in diabetics.

Part 2.2. Role of society

I do not quite understand what this phrase "frustrated by a dominated virtual environment" means.

I suggest that the authors provide a brief introduction on the NAFLD/MASLD SDG framework, and include details or some examples on how these identified SDG indicators impact MASLD and can be used in the multimodal treatment of MASLD (e.g., governments and policymakers, lobby and advocacy groups, etc.).

Paragraph 2: Kindly provide more detail on these MASLD-oriented digital therapeutics and internet applications.

Part 2.3. Psychiatric comorbidity

Are there studies/data on the efficacy of interventions such as CBT in facilitating lifestyle changes and treating MASLD and other metabolic conditions?

Since some psychiatric pharmacotherapies are associated with weight gain, what are the authors' suggestions on how to address this issue in patients with MASLD and psychiatric comorbidities, who are concomitantly also counseled to lose weight?

Part 3.2. Education on prognosis

Educating patients that steatosis and steatohepatitis are reversible, and that fibrosis can be stabilized (?regressed), through lifestyle changes and weight loss can additionally motivate them to commit to lifestyle changes. Including data on steatosis and MASH resolution that are available from clinical trials on lifestyle interventions would be helpful in this regard.

Part 4.1. Socioeconomic status and education

I think it is important to include data and evidence on the association of food security and other socioeconomic metrics with MASLD and outcomes. It might also be illustrative to mention and describe the concepts of “food deserts” and “food swamps” since many low-income patients unfortunately live in these kinds of neighborhoods.

Low socioeconomic status also limits access to an exercise routine. Gym memberships cost money, some communities are lacking in social and recreational programs, and some patients simply have little or no time left after their jobs and family responsibilities.

Part 5. I am not sure “What are the expectations?” best describes this section. Perhaps the authors should clarify what they mean by “expectations”.

Part 6. Patient Engagement

It is also important to emphasize that lifestyle modifications and weight loss have broader benefits that extend beyond the liver, including improvements in metabolic and cardiovascular health and cancer prevention, while pharmacotherapies often have a narrower scope of efficacy (with the exception perhaps of GLP-1 and GIP agonists).

Reviewer #2

(Remarks to the Author)

In this manuscript, Boeckmans et. al. aim to provide strategies to engage patients with MASLD to adopt lifestyle changes and improve their health. They highlight the importance of the physician patient relationship in successful treatment of MASLD and provide an in-depth literature discuss strategies to build trustful relationships with patients and provide examples of how to align with patients to incorporate lifestyle changes and improve liver health.

Major Comments:

1. It is not clear who is the target audience. Are the authors targeting a general practice provider or gastroenterologist/hepatologist. Defining the target audience would further clarify the average patient population of that specific provider. The focus and pointers of the manuscript appears to be a general approach of managing obesity and cardiometabolic disease which while critical to the treatment of MASLD may not be the focus of a subspecialist. There are not significant data/pointers for the management of MASLD or recommendations specifically for patients with advanced fibrosis.
2. The authors cite a significant amount of compelling data regarding the utility and need for multi-disciplinary teams in addressing MASLD. Nonetheless, the authors do not share any data or present a model of how these different subspecialties should integrate or suggest any triage process for which patient would benefit from which services. As it is not realistic for every patient to be evaluated by all the subspecialty care specialists, perhaps the authors could provide a diagram or lay out their vision of what such a subspecialty clinic would look like. As the goal is to provide the patient with the best care possible limiting unnecessary evaluations would similarly be valuable.
3. The forward looking statemen section 7.2 seems to be largely a recap of the remainder of the manuscript similar to section 7.1. Perhaps these sections can be combined, or be used to provide a vision of the future of MASLD care without reviewing the prior literature.
4. Consider discussing the role and influence of a patient’s support system in their health, particularly in section 2, Navigating in a relationship of trust. Building rapport and involving close family and/or friends in a patient’s care can be incredibly helpful, particularly for those who may have more advanced disease and dependance of family.
5. The manuscript should be reviewed carefully for clarity. The message of the manuscript would likely benefit from being slightly condensed with more straightforward sentence structure.

Minor Comments:

1. Figure 3 – Consider making the text at an angle as it is difficult to read. Please identify the significance of the different arrow colors.

Reviewer #3

(Remarks to the Author)

The paper is relevant and important with regard to patient care with those whom have been diagnosed with MASLD and MASH, and for anyone who has yet to identified as a patient. What struck me was the overall encompassing theme was the focus on a tailored care approach, an d one that builds trust and rapport between the Physician and the patient. Every patient is unique and impacted by a myriad of individual factors, including their ethnicity, their financial status, their geographical location, their available behavioral support, their accessibility to patient supportive tools and more.

There was also discussion about the importance of the physician and their ability and interest in taking the time to understand each patient, to explain the results of their liver assessments, and careful, thoughtful consideration about how best to communicate to each patient so that they would be motivated to make positive changes in their health. The paper includes comments about food and smoking and alcohol which helps ensure that these important factors are included and considered.

The paper would be of interest and helpful, especially for Primary Care Physicians as well as others related in the multi-disciplinary care of MASLD/MASH patients.

I did find a number of typos and grammatical errors:

Lines:

- 77 - unravelling versus unravel
- 121 - patients versus patient
- 150 - delete the word on
- 172 - delete each and change to all
- 173 - change patient to patients and the word they to it.
- 174 - after the word explained add in 'to them'.
- 191 - change need to needs
- 192 - change the word on to 'with'
- 204 - change the word for to 'of'
- 226 - I don't think the word 'anamnestic' is appropriate
- 234 - I don't think the word 'anamnestic' is appropriate
- 240 - change easy to 'easily'
- 241 - change lays to 'lies'
- 268 - add the word 'have' between that and ever
- 280 - should be results in 'a' lack of basic knowledge 'about' the disease
- 295 - delete 'also'
- 296 - change to can also be used
- 316 - change from attained to attain
- 338 - change from 'may not go at' to 'may be adopted at'
- 351 - change from 'made from own' to made from 'the Physician's own'
- 377 - correct 'diner' to 'dinner'
- 405 - correct 'setting' to 'set' and working to work
- 416 - correct 'individual' to 'individual'
- 426 - correct 'easier' to 'more easily'
- 430 - change stadalone to 'stand alone'
- 439 - delete the word 'often'
- 463 - add in the word 'their' : 'searching for their own'
- 466 - change to 'knowledge 'about MASLD' and from 'will yet de-stigmatize' to 'may de-stigmatize'
- 467 - change 'on MASLD' to 'about MASLD'
- 472 - change 'patients on their liver' to 'about their liver'
- 475 - change 'MASLD on the' to 'MASLD in the'
- 478 - change 'In the availability' to 'With the availability'
- 513 - change 'also allowing' to 'allow' and 'practitioners evaluating' to practitioners to evaluate'
- 523 - Consider adding 'in the U.S. because India had a MASH drug approved a few years ago.
- 524 - change 'comes to a new' to 'has reached a new'
- 536 - spell correct 'calory' to 'calorie'

Reviewer #4

(Remarks to the Author)

Solid article with that will ka e a solid contribution to the field. A few cut/paste suggestions from the manuscript:

- There are several spelling (e.g., 'administrtion') and grammar errors. Please edit.

- "Emotionally neutral non-invasive tests can assist in this process but require context on their advantages and limitations."
---I'm not sure what 'emotionally neutral' means here. In the structure, it means the test/data otherwise has an emotion. Doesn't seem necessary. Similar in other mentions of 'emotionally neutral.' At best it is awkward. At worst, it is patronizing and unnecessary.

- "Discussing and setting lifestyle goals is a mandatory (DELETE 'mandatory', insert 'priority') before initiating MASLD-

specific pharmacological treatment since living a healthy lifestyle will remain the basis of the multimodal management of MASLD."

---Otherwise, this statement could be seen in support of step therapy for patients with fibrosis.

- "On the other hand, creating a safe environment and providing statements such as 'we will make a plan together to improve your liver condition, which will put you on the rails (DELETE 'the rails', insert 'track') for a healthy life' can sound (DELETE 'sound', insert 'be') motivational and give a sense of team spirit (DELETE 'team spirit', insert 'provider-patient partnership' to strive for a better quality of life..."

---'On the rails' has a negative meaning associated with last chance. eg., barely hanging on. Other suggestions move from subjective, which puts emphasis on patient to more objective relationship process between patient and provider.

- "With the clear (delete 'clear', insert 'potential') stigmatizing effects of obesity and MASLD 9, the use of emotionally neutral NITs can at least partly eliminate the experience of self-blaming and provide neutral targets when adopting lifestyle modifications."

----Not sure I see the purpose of this sentence as it relates to the advantages of NITs vs biopsy. Both establish "emotionally neutral" data.

- "As a result, one might question whether certain metabolic goals obtained through effective patient communication and multimodal treatment should be achieved, or at least attempted, before specific drug treatment can be initiated."

---This statement could be used as a justification for step therapy - and delay of medication - for those needing available, immediate fibrotic medication.

Version 1:

Reviewer comments:

Reviewer #1

(Remarks to the Author)

I thank the authors' receptiveness to the reviewers' comments. I am pleased with the authors' modifications to the manuscript, which addressed reviewer comments (both mine and others) but maintained the original tenor of their paper. I recommend acceptance.

Reviewer #2

(Remarks to the Author)

Thank you to the authors for addressing the comments.

No additional comments at this time

Please note that the heading to section 2.1 reads "Emphatic listening" I presume the authors mean to write "Empathetic listening"

Reviewer #3

(Remarks to the Author)

Reviewer #4

(Remarks to the Author)

This is a helpful start for engaging the many issues around the relationship between individual behavior and environment and metabolic liver health. I am inspired to see this being addressed from within the liver community in place of experts from other fields weighing in.

In my comments I note a few concerns and issues with language. Although not highlighted in my review, the article would benefit from a grammar check before publication. Thanks again for the opportunity. This is an excellent conversation for the field to engage.

1. Abstract [98/100 words]

Metabolic dysfunction-associated steatotic liver disease (MASLD) is often regarded as a disease caused by personal dietary and lifestyle choices.

- This is vague and ignores genetic, environmental, or co-occurring disorder's influence on disease occurrence and/or progression. Regarded by whom -Society? Providers? Payors? Patients?

2. Societal factors contribute to the increasing prevalence of MASLD by promoting the development of these metabolic disorders through the facilitation of a sedentary lifestyle and stimulation of ultra-processed food consumption (Fig. 1)2,14

- Clarifying examples of what "societal factors" may be helpful. "Societal" is so big a reference as to be almost meaningless.

I would encourage the authors to include references such as "e.g. commercial, environmental."

3. Policy makers and the food industry hence have important roles in controlling MASLD in the population, although these are often affected by financial incentives.

- A mention of environmental aspects could be clarifying. While policymakers and the food industry are important factors, low-income communities without access to easy, safe physical activity options are important. This could include sidewalk access, green space, and well-lit public areas.

4. The most readily modifiable factors thus lay in personal behavior, including having a healthy diet and regular physical activity, and avoiding tobacco and alcohol use¹⁵⁻¹⁷

- Add "Depending on the stage of MASLD/MASH and comorbidities..."

5. In addition, patients with MASLD have limited readiness to adopt lifestyle changes, especially regarding exercise¹⁹...

- Great and needed statement!

6. 2.2 Establishing mutual goals and identifying personal factors that hinder change to engage the patient to collaborate with other experts

- Should this not be mutual "strategies" with the objective goal of resolving liver disease?

- I am concerned with the clinician identifying personal matters that may not be in the area of their expertise - such as nutrition or physical activity. Cultural competency is important also as referencing a Mediterranean diet is often not understood or practical.

7. Ideally, this process results in intuitive engagement to collaborating...

- Not intuitive. It should be intentional.

8. ...leading to multimodal treatment of MASLD and associated diseases²⁹.

- Great recommendation.

9. ...these are non-linear achievements of the goal and integral elements of their process, and that changing lifestyle constitutes a matter of learning and reflecting in which one can improve (Fig. 3)

- Well said.

10. In addition, discussing the differences between body weight and body composition³⁷ is crucial to patients improving their health through physical exercise without achieving weight loss.

- This is such an important point that it would be good to see emphasized earlier and in more detail. Often by using "weight loss" - we are referring to "fat loss", especially for liver-related conditions. Unfortunately, the history and body of research refer to weight loss. Clarification and better understanding could lead to patient improvement.

11. ...patients feeling stigmatized and guilty about their metabolic condition and potentially frustrated by a virtual environment portraying a beauty ideal.

- It's not always based in the vanity of attractiveness - it's also youth, elite athleticism, sexuality, and misguided appearance of health. Also, it's more than a virtual environment. There are very real environmental factors that play a part.

12. ...urban green space indicator...

- Thank you for this important inclusion.

13. ...can yield patients the realization that they should not blame themselves

- I would reframe from the negative to a more positive - "that there are great forces working against the individual struggling for better health outcomes." Reducing stigma includes not presuming there is stigma to begin with.

14. the generic sports advice

- Not sure what this term means.

15. ...behavioral activation therapy as the initial starting point...

- Not sure behavioral activation therapy is the starting point for mental health care. It may require more specialized psychological/psychiatric methods.

16. Apart from depression, there also exists a relationship between MASLD and the development of anxiety disorders...

- Happy to see this included as these conditions are often different manifestations of a similar root. They definitely can co-exist.

17. To these patients, emotional support and creating a safe space for sharing experiences and fears are even more important.

- "To all patients...."

18. 3. Engagement by information

NAFLD, MAFLD, or MASLD: the forgotten or yet to discover non-communicable disease?

With NAFLD, metabolic dysfunction-associated fatty liver disease (MAFLD), and MASLD, three different acronyms have been used in recent years to describe in essence the same disease entity⁵⁸.

- Not sure this section is necessary in the context of the other information in the article. As MAFLD is now the term of use in some countries, it would be positive to emphasize the shift from 'non-alcoholic' to 'metabolic' as that is the common factor across all designations.

19. Educational material, for example from EASL70...

- Broaden this to "from medical associations (such as EASL) and patient organizations"

20. Another initiative to inform patients with MASLD is the Global Fatty Liver Day, which is a public education campaign supported by multiple medical societies71.

- include supported by "liver patient organizations and multiple..."

21. although improvement in these tests over time might be further motivational.

- There is a consistent use of "motivational" in the article's language without direction as to who it would be motivating for. Is it implied the patient needs motivation - or would these figures be clarifying and empowering to the already motivated patient?

22. low socio-economic status may also limit access to sports facilities and exercise routine because of costly memberships or time constraints92.

- consider "physical activity options" in place of "sports facilities."

23. This new disease category not only allows better classification of patients, but its use can also motivate patients to limit their alcohol consumption.

- Great inclusion.

24. A first important step in avoiding or limiting alcohol use is making patients aware of their drinking habits.

- Not sure talking about the individual behavior is the correct first step. That can easily be perceived as a 'blame the patient' approach. First step could also be depersonalized by talking about negative health outcomes, specific impact on the liver, etc.

25. One may, however, not forget the basic concept of patient-centered communication and minimally disruptive care and should seek which expectations patients have and what level of care they prefer119,120.

- Excellent.

26. If patients' intuitive engagement is thought to be insufficient...

- The term 'Intuitive' appears many times in the article too. Not sure what that means in the clinical approach.

27. one should not be afraid to acknowledge the emotional and structural factors that might underlie their inadequate motivation.

- As noted previously, pre-supposing "inadequate motivation" may be misplaced when the patient is motivated but needs better support, education, or empowerment for understanding.

28. Nonetheless, relying solely on primary care to bring a strong message to patients with MASLD and initiate a holistic treatment plan involving a tailored multidisciplinary team is not feasible due to time constraints and specialized aspects of follow-up.

- Great point.

29. In addition, we expect policymakers to actively promote disease destigmatization while limiting promotion and advertising of calorie-rich and highly processed foods.

- This conclusion statement seems awkwardly phrased. "Calorie rich" could also be "nutrient dense" and thus good. The connection between promotion and advertising is not otherwise explored in the article and could raise issues (e.g., protected commercial speech, regulatory definitions of "healthy") not directly connected to liver health outcomes.

Not sure the final sentence is needed. The intent is correct but the language should be avoided or clarified.

Version 2:

Reviewer comments:

Reviewer #4

(Remarks to the Author)

No comments or additional edits. Modifications and edits are appreciated. Reasoning and explanations in other areas are well understood. Thank you for your openness to feedback.

Responses to Reviewers

Response to **Reviewer #1** pages **2 - 6**

Response to **Reviewer #2** pages **7 - 8**

Response to **Reviewer #3** pages **9 - 10**

Response to **Reviewer #4** pages **11 - 12**

Response to Reviewer #1

It was a pleasure reviewing the manuscript entitled “Patient Engagement in the Multimodal Management of MASLD” by Boeckmans et al.

I think we can all agree that there is a wealth of ever-growing literature on healthy lifestyle modification strategies and pharmacotherapeutics to treat MASLD. Clinical trial data and real-world clinical experience also show that only some patients succeed in losing the recommended weight to achieve the desired endpoints of steatohepatitis resolution and hepatic fibrosis stabilization.

What seems to be missing, in my opinion, are papers outlining potential factors affecting patients’ successes and failures in changing their lifestyles to improve metabolic abnormalities, including MASLD, and achieve optimal metabolic health. I believe that this incisive summary by Boeckmans, et al on patient engagement, in the MASLD population specifically, bridges this knowledge gap and provides quite practical and valuable guidance for the clinician taking care of MASLD and other metabolic conditions.

We thank the reviewer for their time and effort in improving this paper, and their encouraging comments.

The following are my comments:

Part 2.1. Process of success and failure

I suggest that they divide this section into separate paragraphs outlining their points to organize the points better. For example, one paragraph is on emphatic listening and building mutual trust. Another on establishing mutual goals and plans and identifying personal factors that hinder change to engage the patient to collaborate with other experts. Another on constructive approaches to failure, providing meaningful feedback, and framing the importance of small successes.

We have subdivided the part on success and failure:

2.1. Emphatic listening and building mutual trust

2.2. Establishing mutual goals and identifying personal factors that hinder change to engage the patient to collaborate with other experts

2.3. Constructive approaches to failure to adopt lifestyle changes

Feedback on success can also be seen through changes in liver chemistries, liver stiffness and CAP score, lipids, and glycemic control in diabetics.

We have expanded to section of NITs with parameters that often more readily change compared to liver-related scores, while they also provide information on metabolic health. We added following text: “Although NITs specifically designed for liver disease can assist in patient participation and information, one should keep in mind that changes in these parameters can lag behind more commonly used laboratory measurements of metabolic health. Therefore, earlier changes in for example blood pressure, cholesterol, triglycerides, and HbA1c can be used for achieving shorter-term goals while also working towards metabolic health in general (Anstee et al,; Targher et al.).

References:

-Anstee, Q. M. et al. Prognostic utility of Fibrosis-4 Index for risk of subsequent liver and cardiovascular events, and all-cause mortality in individuals with obesity and/or type 2 diabetes: a longitudinal cohort study. Lancet Reg Health Eur. 36, 100780 (2024).

-Targher, G., Byrne, C. D. & Tilg, H. MASLD: a systemic metabolic disorder with cardiovascular and malignant complications. *Gut*. 73, 691–702 (2024)

Part 2.2. Role of society

I do not quite understand what this phrase “frustrated by a dominated virtual environment” means.

To improve clarity, we have amended the sentence to “Although multimodal treatment of MASLD evidently seems a good approach, one needs to be cautious with patients feeling stigmatized and guilty about their metabolic condition and potentially frustrated by a virtual environment portraying a beauty ideal.”

I suggest that the authors provide a brief introduction on the NAFLD/MASLD SDG framework, and include details or some examples on how these identified SDG indicators impact MASLD and can be used in the multimodal treatment of MASLD (e.g., governments and policymakers, lobby and advocacy groups, etc.).

We have expanded the section of the Sustainable Development Goal score for MASLD and added an example on how to use it: “To that end, a Sustainable Development Goal score has been developed as an advocacy tool for NCDs, including MASLD. The Sustainable Development Goal score for MASLD provides an estimate of the country-level preparedness to manage MASLD from a societal perspective, which can facilitate multisectoral collaborations. Indicators for sustainable development regarding MASLD are child wasting, child overweight, NCD mortality, a universal health coverage service coverage index, health worker density, education attainment, and as well an urban green space indicator, which is important for physical and mental health. Each of these factors can negatively or positively influence the development of MASLD, invigorating the role of society, education, and upbringing. For example, mortality due to diabetes type 2 and cardiovascular disease might not have been health priorities in low- and middle-income countries that instead often rather focus on communicable diseases, which is captured by the Sustainable Development Goal score to subsequently inform policy makers to take action at the population level”

Reference:

-Lazarus, J. V. et al. The global fatty liver disease Sustainable Development Goal country score for 195 countries and territories. *Hepatology*. 78, 911–928 (2023).

Paragraph 2: Kindly provide more detail on these MASLD-oriented digital therapeutics and internet applications.

We have provided more detail on the digital therapeutics: “Digital therapeutics including internet applications specifically designed for patients with MASLD can be proposed to assist in this process. Such applications are tailored to the individual’s needs and assess motivation to change and consciousness on their disease. Further, they provide education using interactive slides and can guide physical exercise (Mazzotti et al.; Pfirrmann et al.) .”

References:

-Mazzotti, A. et al. An internet-based approach for lifestyle changes in patients with NAFLD: Two-year effects on weight loss and surrogate markers. *J Hepatol*. 69, 1155–1163 (2018).

-Pfirrmann, D. et al. Web-Based Exercise as an Effective Complementary Treatment for Patients With Nonalcoholic Fatty Liver Disease: Intervention Study. J Med Internet Res. 21, 211250 (2019).

Part 2.3. Psychiatric comorbidity

Are there studies/data on the efficacy of interventions such as CBT in facilitating lifestyle changes and treating MASLD and other metabolic conditions?

We thank the reviewer for this valuable comment. We have expanded the section on cognitive behavioral therapy and described a meta-analysis investigating the effects of CBT on weight loss. We added following text: “There exist, for example, diverse needs for behavioral support for successful lifestyle change which could range from peer support to coaching to structured psychology and psychiatry needs both at the inter-individual and intra-individual level over time (Uphoff et al.; Allen et al.). Moreover, a meta-analysis investigating the effect of cognitive behavioral therapy on lifestyle changes found that it improves weight loss (effect size (BMI) -0.63 with 95%CI -1.17;-0.10) and weight maintenance effect size (BMI) -0.55 with 95%CI -0.90;-0.20) (Kurnik Mesarič et al.), which is key to treating MASLD (Vilar-Gomez et al.).”

References:

-Uphoff, E. et al. Behavioural activation therapy for depression in adults with non-communicable diseases. Cochrane Database of Syst Rev. 8, CD013461 (2020).

-Allen, L. et al. Socioeconomic status and non-communicable disease behavioural risk factors in low-income and lower-middle-income countries: a systematic review. Lancet Glob Health. 5, e277–e289 (2017).

-Kurnik Mesarič, K., Pajek, J., Logar Zakrajšek, B., Bogataj, Š. & Kodrič, J. Cognitive behavioral therapy for lifestyle changes in patients with obesity and type 2 diabetes: a systematic review and meta-analysis. Sci Rep. 13, 12793 (2023).

-Vilar-Gomez, E. et al. Weight loss through lifestyle modification significantly reduces features of nonalcoholic steatohepatitis. Gastroenterology. 149, 367–378 (2015).

Since some psychiatric pharmacotherapies are associated with weight gain, what are the authors' suggestions on how to address this issue in patients with MASLD and psychiatric comorbidities, who are concomitantly also counseled to lose weight?

We have added a paragraph on the appetite-promoting and weight-inducing effects of different antidepressants, since the section on psychiatric comorbidities mainly targets this disease: “Furthermore, several antidepressants and other drugs used for psychiatric diseases are appetite-promoting and induce weight gain (Hasnain et al.), which can result in a vicious circle. It is the shared responsibility of the prescribing physician and pharmacist to select the least weight-inducing agent for a specific patient, and to inform the patient on this potential side-effect, in particular when the patient is overweight or obese. A potential future avenue consists of the additional prescription of a GLP-1 agonist to pro-actively address expected weight gain induced by psychiatric drugs (Gonzalez et al.; Bak et al.)”

Reference:

-Hasnain, M. & Vieweg, W. V. R. Weight considerations in psychotropic drug prescribing and switching. Postgrad Med. 125, 117–129 (2013).

-Gonzalez, CL., Azim, S., & Miedlich SU. GLP-1 Analogs Are Superior in Mediating Weight Loss But Not Glycemic Control in Diabetic Patients on Antidepressant

Medications: A Retrospective Cohort Study. Prim Care Companion CNS Disord. 9;23:20m02868 (2021).

-Bak M., et al. Glucagon-like peptide agonists for weight management in antipsychotic-induced weight gain: A systematic review and meta-analysis. Acta Psychiatr Scand. In press (2024).

Part 3.2. Education on prognosis

Educating patients that steatosis and steatohepatitis are reversible, and that fibrosis can be stabilized (?regressed), through lifestyle changes and weight loss can additionally motivate them to commit to lifestyle changes. Including data on steatosis and MASH resolution that are available from clinical trials on lifestyle interventions would be helpful in this regard.

We have amended and expanded the paragraph on education: “Therefore, education on the natural history of MASLD and the reversibility of liver steatosis, MASH and fibrosis through adopting lifestyle changes are key aspects to promoting intuitive motivation and preventing the terminal complications of MASLD (Vilar-Gomez et al.; Hagström et al.)” And “Yet, awareness about having liver fibrosis can potentially already promote a healthier lifestyle, since unawareness about liver fibrosis stage by people with MASLD/MASH has been shown to be associated with poor adherence to lifestyle changes (Carrieri et al.)”.

References:

-Vilar-Gomez, E. et al. Weight loss through lifestyle modification significantly reduces features of nonalcoholic steatohepatitis. Gastroenterology. 149, 367–378 (2015).

-Hagström, H., Shang, Y., Hegmar, H., Nasr, P. Natural History and Progression of Metabolic Dysfunction-Associated Steatotic Liver Disease. Lancet Gastroenterol Hepatol. 9, 944-956 (2024).

-Carrieri, P. et al. Knowledge of liver fibrosis stage among adults with NAFLD/NASH improves adherence to lifestyle changes. Liver Int. 42, 984–994 (2022).

Part 4.1. Socioeconomic status and education

I think it is important to include data and evidence on the association of food security and other socioeconomic metrics with MASLD and outcomes. It might also be illustrative to mention and describe the concepts of “food deserts” and “food swamps” since many low-income patients unfortunately live in these kinds of neighborhoods.

We have added following paragraph on food insecurity: “Further, a low socio-economic status goes hand in hand with food insecurity, which is a risk factor for MASLD. In this regard, ‘food deserts’, areas with sparse options to acquire nutritious food, and ‘food swamps’, areas with a high concentration of fast food- and junk food-selling restaurants, create an obesogenic climate that promote the development and worsening of MASLD (Zelber-Sagi et al.)”.

Reference:

-Zelber-Sagi, S. et al. Food inequity and insecurity and MASLD: burden, challenges, and interventions. Nat Rev Gastroenterol Hepatol. 21, 668-686 (2024).

Low socioeconomic status also limits access to an exercise routine. Gym memberships cost money, some communities are lacking in social and recreational programs, and some patients simply have little or no time left after their jobs and family responsibilities.

We have added following sentence to the section on socio-economic status: “In line with this, low socio-economic status may also limit access to sports facilities and exercise routine because of costly memberships or time constraints (Richard et al.).”

Reference:

Richard, V. et al. Socioeconomic inequalities in sport participation: pattern per sport and time trends – a repeated cross-sectional study. BMC Public Health. 23, 785 (2023).

Part 5. I am not sure “What are the expectations?” best describes this section. Perhaps the authors should clarify what they mean by “expectations”.

Because it can be confusing for readers to know whose expectations we are referring to, we changed the title to: 5. “What are the patient’s expectations and preferences?”

Part 6. Patient Engagement

It is also important to emphasize that lifestyle modifications and weight loss have broader benefits that extend beyond the liver, including improvements in metabolic and cardiovascular health and cancer prevention, while pharmacotherapies often have a narrower scope of efficacy (with the exception perhaps of GLP-1 and GIP agonists).

We emphasized on the broader benefits of adopting lifestyle changes: “Nevertheless, lifestyle changes will remain essential to achieve holistic better metabolic health, including cardiovascular benefits, and avoid sarcopenia.”

Response to Reviewer #2

In this manuscript, Boeckmans et. al. aim to provide strategies to engage patients with MASLD to adopt lifestyle changes and improve their health. They highlight the importance of the physician patient relationship in successful treatment of MASLD and provide an in-depth literature discuss strategies to build trustful relationships with patients and provide examples of how to align with patients to incorporate lifestyle changes and improve liver health. **We thank the reviewer for their time and effort in improving this paper. The raised comments are relevant, and we have reworked our manuscript accordingly.**

Major Comments:

1. It is not clear who is the target audience. Are the authors targeting a general practice provider or gastroenterologist/hepatologist. Defining the target audience would further clarify the average patient population of that specific provider. The focus and pointers of the manuscript appears to be a general approach of managing obesity and cardiometabolic disease which while critical to the treatment of MASLD may not be the focus of a subspecialist. There are not significant data/pointers for the management of MASLD or recommendations specifically for patients with advanced fibrosis.

We have specified the target audience at the end of the introduction and delineated the patient population: “This practical guide for the hepatologist and allied healthcare workers aims at providing concrete tips to engage patients with non-cirrhotic MASLD to adopting lifestyle changes and improve their liver condition and overall metabolic health.”

2. The authors cite a significant amount of compelling data regarding the utility and need for multi-disciplinary teams in addressing MASLD. Nonetheless, the authors do not share any data or present a model of how these different subspecialties should integrate or suggest any triage process for which patient would benefit from which services. As it is not realistic for every patient to be evaluated by all the subspecialty care specialists, perhaps the authors could provide a diagram or lay out their vision of what such a subspecialty clinic would look like. As the goal is to provide the patient with the best care possible limiting unnecessary evaluations would similarly be valuable.

We have added a practical way on how the collaborations between the different subspecialties could be coordinated in daily life: “From a practical point of view, the central persons coordinating these partnerships would be the hepatologist and nurse in cases of more advanced disease, and the primary care provider for patients with early stage MASLD. Monthly or quarterly multidisciplinary meetings could be organized to evaluate the goals and needs of the individual patients.”

3. The forward looking statemen section 7.2 seems to be largely a recap of the remainder of the manuscript similar to section 7.1. Perhaps these sections can be combined, or be used to provide a vision of the future of MASLD care without reviewing the prior literature.

We have combined the conclusion and forward-looking statement section and condensed the information.

4. Consider discussing the role and influence of a patient’s support system in their health, particularly in section 2, Navigating in a relationship of trust. Building rapport and involving

close family and/or friends in a patient's care can be incredibly helpful, particularly for those who may have more advanced disease and dependence on family.

We have added following sentence to the paragraph on building mutual trust: "Family or friends accompanying the patient can be involved in this process to assist in a supportive network on a daily basis (Ho et al.)."

Reference:

-Ho, Y. C. L., Mahirah, D., Zhong-Hao Ho, C., Thumboo, J. The role of the family in health promotion: a scoping review of models and mechanisms. *Health Promot Int.* 37, 1-14 (2022).

5. The manuscript should be reviewed carefully for clarity. The message of the manuscript would likely benefit from being slightly condensed with more straightforward sentence structure.

We have condensed the manuscript and removed the numeric values from table 1 to bring a clearer message.

Minor Comments:

1. Figure 3 – Consider making the text at an angle as it is difficult to read. Please identify the significance of the different arrow colors.

We have made the text at an angle and added a color legend to the figure.

Response to Reviewer #3

The paper is relevant and important with regard to patient care with those whom have been diagnosed with MASLD and MASH, and for anyone who has yet to identified as a patient. What struck me was the overall encompassing theme was the focus on a tailored care approach, and one that builds trust and rapport between the Physician and the patient. Every patient is unique and impacted by a myriad of individual factors, including their ethnicity, their financial status, their geographical location, their available behavioral support, their accessibility to patient supportive tools and more.

There was also discussion about the importance of the physician and their ability and interest in taking the time to understand each patient, to explain the results of their liver assessments, and careful, thoughtful consideration about how best to communicate to each patient so that they would be motivated to make positive changes in their health. The paper includes comments about food and smoking and alcohol which helps ensure that these important factors are included and considered.

The paper would be of interest and helpful, especially for Primary Care Physicians as well as others related in the multi-disciplinary care of MASLD/MASH patients.

I did find a number of typos and grammatical errors:

Lines:

77 - unravelling versus unravel

121 - patients versus patient

150 - delete the word on

172 - delete each and change to all

173 - change patient to patients and the word they to it.

174 - after the word explained add in 'to them'.

191 - change need to needs

192 - change the word on to 'with'

204 - change the word for to 'of'

226 - I don't think the word 'anamnestic' is appropriate

234 - I don't think the word 'anamnestic' is appropriate

240 - change easy to 'easily'

241 - change lays to 'lies'

268 - add the word 'have' between that and ever

280 - should be results in 'a' lack of basic knowledge 'about' the disease

295 - delete 'also'

296 - change to can also be used

316 - change from attained to attain

338 - change from 'may not go at' to 'may be adopted at'

351 - change from 'made from own' to made from 'the Physician's own'

377 - correct 'diner' to 'dinner'

405 - correct 'setting' to 'set' and working to work

416 - correct 'indivudual' to 'individual'

426 - correct 'easier' to 'more easily'

430 - change stadalone to 'stand alone'

- 439 - delete the word 'often'
- 463 - add in the word 'their' : 'searching for their own'
- 466 - change to 'knowledge 'about MASLD' and from 'will yet de-stigmatize' to 'may de-stigmatize'
- 467 - change 'on MASLD' to 'about MASLD'
- 472 - change 'patients on their liver' to 'about their liver'
- 475 - change 'MASLD on the' to 'MASLD in the'
- 478 - change 'In the availability' to 'With the availability'
- 513 - change 'also allowing' to 'allow' and 'practitioners evaluating' to 'practitioners to evaluate'
- 523 - Consider adding 'in the U.S. because India had a MASH drug approved a few years ago.'
- 524 - change 'comes to a new' to 'has reached a new'
- 536 - spell correct 'calory' to 'calorie'

We thank the reviewer for their time and effort in improving this paper. The identified typos and grammar errors are relevant, and we have corrected these as suggested.

Response to Reviewer #4

Solid article with that will be a solid contribution to the field. A few cut/paste suggestions from the manuscript:

We thank the reviewer for their time and effort in improving this paper, and their encouraging words.

- There are several spelling (e.g., 'administrtion') and grammar errors. Please edit. **Please see reply to reviewer #3 as well. We have corrected the typos and grammatical errors.**

- "Emotionally neutral non-invasive tests can assist in this process but require context on their advantages and limitations."

---I'm not sure what 'emotionally neutral' means here. In the structure, it means the test/data otherwise has an emotion. Doesn't seem necessary. Similar in other mentions of 'emotionally neutral.' At best it is awkward. At worst, it is patronizing and unnecessary.

We have removed the term 'emotionally-neutral' throughout the manuscript or replaced it by 'objective'.

- "Discussing and setting lifestyle goals is a mandatory (DELETE 'mandatory', insert 'priority') before initiating MASLD-specific pharmacological treatment since living a healthy lifestyle will remain the basis of the multimodal management of MASLD."

---Otherwise, this statement could be seen in support of step therapy for patients with fibrosis.

We have replaced 'mandatory' by 'priority' throughout the manuscript.

- "On the other hand, creating a safe environment and providing statements such as 'we will make a plan together to improve your liver condition, which will put you on the rails (DELETE 'the rails', insert 'track') for a healthy life' can sound (DELETE 'sound', insert 'be') motivational and give a sense of team spirit (DELETE 'team spirit', insert 'provider-patient partnership' to strive for a better quality of life..."

---'On the rails' has a negative meaning associated with last chance. eg., barely hanging on. Other suggestions move from subjective, which puts emphasis on patient to more objective relationship process between patient and provider.

We have amended the text accordingly: "On the other hand, creating a safe environment and providing statements such as 'we will make a plan together to improve your liver condition, which will put you on track for a healthy life' can be motivational and give a sense of provider-patient partnership to strive for a better quality of life and reduced risk of both liver- and non-liver-related outcomes (Figure 2)."

- "With the clear (delete 'clear', insert 'potential') stigmatizing effects of obesity and MASLD 9, the use of emotionally neutral NITs can at least partly eliminate the experience of self-blaming and provide neutral targets when adopting lifestyle modifications."

----Not sure I see the purpose of this sentence as it relates to the advantages of NITs vs biopsy. Both establish "emotionally neutral" data.

We have updated the sentence to "With the potential stigmatizing effects of obesity and MASLD, the use of NITs can at least partly eliminate the experience of self-blaming and provide objective targets when adopting lifestyle modifications."

- "As a result, one might question whether certain metabolic goals obtained through effective patient communication and multimodal treatment should be achieved, or at least attempted, before specific drug treatment can be initiated."

---This statement could be used as a justification for step therapy - and delay of medication - for those needing available, immediate fibrotic medication.

We have amended the paragraph to "As a result, one might question whether certain metabolic goals obtained through patient engagement should be achieved, or at least attempted, before specific drug treatment can be initiated. In that view, adopting lifestyle modifications can be used as a justification for pharmacological treatment in a second step."

Response to Reviewers

Author's correspondence to the reviewers: We appreciate the recommendation from Reviewer #1 and #2 to accept our manuscript for publication in *Communications Medicine* and gratefully implemented the valuable input from Reviewer #4 to further improve the manuscript. Please find below specific responses to the raised points.

Reviewer #1:

I thank the authors' receptiveness to the reviewers' comments. I am pleased with the authors' modifications to the manuscript, which addressed reviewer comments (both mine and others) but maintained the original tenor of their paper. I recommend acceptance.

Author's reply: Thank you for your appreciation of our work and recommendation for acceptance.

Reviewer #2:

Thank you to the authors for addressing the comments.

No additional comments currently

Please note that the heading to section 2.1 reads "Emphatic listening" I presume the authors mean to write "Empathetic listening"

Author's reply: Thank you for your appreciation of our work and bringing this typo to our attention, which has been resolved.

Reviewer #4:

This is a helpful start for engaging the many issues around the relationship between individual behavior and environment and metabolic liver health. I am inspired to see this being addressed from within the liver community in place of experts from other fields weighing in.

In my comments I note a few concerns and issues with language. Although not highlighted in my review, the article would benefit from a grammar check before publication. Thanks again for the opportunity. This is an excellent conversation for the field to engage.

Thank you for your appreciation of our work and highlighting these points to improve our manuscript. We did an additional grammar check and we feel that your input has strengthened our manuscript.

Comment 1. Abstract [98/100 words]

Metabolic dysfunction-associated steatotic liver disease (MASLD) is often regarded as a disease caused by personal dietary and lifestyle choices.

- This is vague and ignores genetic, environmental, or co-occurring disorder's influence on disease occurrence and/or progression. Regarded by whom -Society? Providers? Payors? Patients?

Author's reply: We have amended the sentence to highlight the role of society: "Metabolic dysfunction-associated steatotic liver disease (MASLD) is often regarded in society as a disease caused by personal lifestyle and dietary choices."

Comment 2. Societal factors contribute to the increasing prevalence of MASLD by promoting the development of these metabolic disorders through the facilitation of a sedentary lifestyle and stimulation of ultra-processed food consumption (Fig. 1)^{2,14}

- Clarifying examples of what "societal factors" may be helpful. "Societal" is so big a reference as to be almost meaningless. I would encourage the authors to include references such as "e.g. commercial, environmental."

Author's reply: Thank you for this comment. We would like to clarify that the different societal factors contributing to MASLD development are mentioned in Figure 1 (i.e. social network, urbanization, finances, food insecurity, lack of information and awareness, psychiatric comorbidity, and access to healthcare). To avoid repetition, we opted to keep the reference to the figure (which will be presented close to this paragraph in the final paper, if accepted) rather than describe it in the text.

Comment 3. Policy makers and the food industry hence have important roles in controlling MASLD in the population, although these are often affected by financial incentives.

- A mention of environmental aspects could be clarifying. While policymakers and the food industry are important factors, low-income communities without access to easy, safe physical activity options are important. This could include sidewalk access, green space, and well-lit public areas.

Author's reply: Access to physical activity options and green space are indeed important factors for a healthy lifestyle. Therefore, we described these aspects in dedicated parts of manuscript, for example: "The Sustainable Development Goal score for MASLD provides an estimate of the country-level preparedness to manage MASLD from a societal perspective, which can facilitate multisectoral collaborations. Indicators for sustainable development regarding MASLD are child wasting, child overweight, NCD mortality, a universal health coverage service coverage index, health worker density, education attainment, and as well an urban green space indicator, which is important for physical and mental health." and "In line with this, low socio-economic status may also limit access to sports facilities and exercise routine because of costly memberships or time constraints."

Comment 4. The most readily modifiable factors thus lay in personal behavior, including having a healthy diet and regular physical activity, and avoiding tobacco and alcohol use¹⁵⁻¹⁷
- Add "Depending on the stage of MASLD/MASH and comorbidities..."

Author's reply: Thank you for this important detail, we have amended the sentence as follows: "The most readily modifiable factors, depending on the stage of MASLD and comorbidities, thus lay in personal behavior, including having a healthy diet and regular physical activity, and avoiding tobacco and alcohol use."

Comment 5. In addition, patients with MASLD have limited readiness to adopt lifestyle changes, especially regarding exercise¹⁹...
- Great and needed statement!

Author's reply: Thank you.

Comment 6. 2.2 Establishing mutual goals and identifying personal factors that hinder change to engage the patient to collaborate with other experts
- Should this not be mutual "strategies" with the objective goal of resolving liver disease?
- I am concerned with the clinician identifying personal matters that may not be in the area of their expertise - such as nutrition or physical activity. Cultural competency is important also as referencing a Mediterranean diet is often not understood or practical.

Author's reply: Thank you for highlighting this nuance. We have amended the sentence as follows: "Establishing mutual strategies and identifying personal factors that hinder change to engage the patient to collaborate with other experts." In regard of cultural competency, we earlier added a paragraph that details aspects of it, for example, but not limited to: "after which culturally tailored adaptations to diet can be made employing foods to which patients are familiar with to result in sustainable behavioral changes."

Comment 7. Ideally, this process results in intuitive engagement to collaborating...
- Not intuitive. It should be intentional.

Author's reply: Thank you for pointing this out. We felt that 'intrinsic' might even be a better word describing the motivation that is required to make sustainable lifestyle changes and therefore replaced 'intuitive' by 'intrinsic' throughout the text.

Comment 8. ...leading to multimodal treatment of MASLD and associated diseases²⁹.
- Great recommendation.

Author's reply: Thank you.

Comment 9. ...these are non-linear achievements of the goal and integral elements of their process, and that changing lifestyle constitutes a matter of learning and reflecting in which one can improve (Fig. 3)

- Well said.

Author's reply: Thank you.

Comment 10. In addition, discussing the differences between body weight and body composition³⁷ is crucial to patients improving their health through physical exercise without achieving weight loss.

- This is such an important point that it would be good to see emphasized earlier and in more detail. Often by using "weight loss" - we are referring to "fat loss", especially for liver-related conditions. Unfortunately, the history and body of research refer to weight loss. Clarification and better understanding could lead to patient improvement.

Author's reply: We strongly agree that body composition is superior to weight, which is why we made this statement. However, this statement appears in the first section after the introduction, which is the earliest way.

Comment 11. ...patients feeling stigmatized and guilty about their metabolic condition and potentially frustrated by a virtual environment portraying a beauty ideal.

- It's not always based in the vanity of attractiveness - it's also youth, elite athleticism, sexuality, and misguided appearance of health. Also, it's more than a virtual environment. There are very real environmental factors that play a part.

Author's reply: Thank you for your help in optimizing this sentence. We have updated the sentence as follows: "Although multimodal treatment of MASLD evidently seems a good approach, one needs to be cautious with patients feeling stigmatized and guilty about their metabolic condition and potentially frustrated by a virtual environment portraying a beauty ideal, elite athleticism, sexuality, and misguided appearance of health."

Comment 12. ...urban green space indicator...

- Thank you for this important inclusion.

Author's reply: Thank you. For your information, this section also refers to the response to comment #3.

Comment 13. ...can yield patients the realization that they should not blame themselves

- I would reframe from the negative to a more positive - "that there are great forces working against the individual struggling for better health outcomes." Reducing stigma includes not presuming there is stigma to begin with.

Author's reply: Thank you for your help in optimizing this sentence, which was updated as follows: "Communicating lifestyle changes through this holistic approach can yield patients the realization that there are forces working against the individual struggling for better health outcomes, and from a societal perspective it can make a patient more aware of lifestyle- and food- related signals in daily life."

Comment 14. the generic sports advice

- Not sure what this term means.

Author's reply: We have replaced 'generic' by 'general' to make the statement clearer.

Comment 15. ...behavioral activation therapy as the initial starting point...

- Not sure behavioral activation therapy is the starting point for mental health care. It may require more specialized psychological/psychiatric methods.

Author's reply: We agree that behavioral activation therapy is not the starting point for mental health care for everyone, which is why this statement should be interpreted in the context of the other parts of the sentence: "Nonetheless, it remains vital to identify details that are suggestive for a clinically relevant or subthreshold depression since it would require involvement specific care whether or not with behavioral activation therapy as the initial starting point, although strong evidence is lacking for such an approach." We therefore argue that this statement does not need a modification.

Comment 16. Apart from depression, there also exists a relationship between MASLD and the development of anxiety disorders...

- Happy to see this included as these conditions are often different manifestations of a similar root. They definitely can co-exist.

Author's reply: Thank you for agreeing on this important topic.

Comment 17. To these patients, emotional support and creating a safe space for sharing experiences and fears are even more important.

- "To all patients...."

Author's reply: We strongly agree that emotional support and a safe space for sharing experiences are important for all patients, which is why this statement should be interpreted in the context of the full sentence: "To these patients, emotional support and creating a safe space for sharing experiences and fears are even more important."

Since this sentence is positioned in the broader scope of psychiatric comorbidity (paragraph 2.5.), we argue that statements made in this part of the manuscript should not be generalized.

Comment 18. 3. Engagement by information

NAFLD, MAFLD, or MASLD: the forgotten or yet to discover non-communicable disease?

With NAFLD, metabolic dysfunction-associated fatty liver disease (MAFLD), and MASLD, three different acronyms have been used in recent years to describe in essence the same disease entity⁵⁸.

- Not sure this section is necessary in the context of the other information in the article. As MAFLD is now the term of use in some countries, it would be positive to emphasize the shift from 'non-alcoholic' to 'metabolic' as that is the common factor across all designations.

Author's reply: Thank you for this comment. We would like to underscore that this section is in place to highlight the role of metabolic dysregulation in the new MASLD nomenclature, for example: "In this nomenclature, the potentially stigmatizing terms 'fatty' and 'alcoholic' were removed, and the role of metabolic dysregulation was highlighted". We therefore believe that this section is essential for the understanding of the manuscript for people that are less involved in the rapid developments of our field.

Comment 19. Educational material, for example from EASL⁷⁰...

- Broaden this to "from medical associations (such as EASL) and patient organizations"

Author's reply: Thank you for your help in optimizing this sentence, which was amended as follows: "Educational material from medical associations (such as EASL) and patient organizations can assist in this process."

Comment 20. Another initiative to inform patients with MASLD is the Global Fatty Liver Day, which is a public education campaign supported by multiple medical societies⁷¹.

- include supported by "liver patient organizations and multiple..."

Author's reply: Thank you for your help in optimizing this sentence, which was amended as follows: "In line with this, an initiative to inform patients with MASLD is the 'Global Fatty Liver Day', which is a public education campaign supported by liver patient organizations and multiple medical societies."⁷¹

Comment 21. although improvement in these tests over time might be further motivational.

- There is a consistent use of "motivational" in the article's language without direction as to who it would be motivating for. Is it implied the patient needs motivation - or would these figures be clarifying and empowering to the already motivated patient?

Author's reply: Thank you for this comment. We agree that an individual with MASLD needs at least some initial motivation to begin with lifestyle changes. However, motivation can fluctuate over time, which we extensively describe, among others regarding Figure 3:

Therefore, we believe that motivation should be supported continuously throughout the process of adopting lifestyle changes, which is why we employed this terminology.

Comment 22. low socio-economic status may also limit access to sports facilities and exercise routine because of costly memberships or time constraints⁹².

- consider "physical activity options" in place of "sports facilities."

Author's reply: Thank you for your help in optimizing this statement. We have replaced “sports facilities” by “physical activity options”: “In line with this, low socio-economic status may also limit access to physical activity options and exercise routine because of costly memberships or time constraints.”

Comment 23. This new disease category not only allows better classification of patients, but its use can also motivate patients to limit their alcohol consumption.

- Great inclusion.

Author's reply: Thank you.

Comment 24. A first important step in avoiding or limiting alcohol use is making patients aware of their drinking habits.

- Not sure talking about the individual behavior is the correct first step. That can easily be perceived as a 'blame the patient' approach. First step could also be depersonalized by talking about negative health outcomes, specific impact on the liver, etc.

Author's reply: Thank you for this relevant input. We have amended the sentence to: "The first important steps in avoiding or limiting alcohol use consist of depersonalizing drinking habits and explaining potential negative liver- and non-liver related health outcomes."

Comment 25. One may, however, not forget the basic concept of patient-centered communication and minimally disruptive care and should seek which expectations patients have and what level of care they prefer^{119,120}.

- Excellent.

Author's reply: Thank you.

Comment 26. If patients' intuitive engagement is thought to be insufficient...

- The term 'Intuitive' appears many times in the article too. Not sure what that means in the clinical approach.

Author's reply: We have replaced 'intuitive' by 'intrinsic' throughout the manuscript. Please see also response to comment #7.

Comment 27. one should not be afraid to acknowledge the emotional and structural factors that might underlie their inadequate motivation.

- As noted previously, pre-supposing "inadequate motivation" may be misplaced when the patient is motivated but needs better support, education, or empowerment for understanding.

Author's reply: Thank you for your help in optimizing this sentence, which has been amended as follows: "If patients' intrinsic engagement is thought to be insufficient to achieve improvements in their liver condition, one should not be afraid to acknowledge the emotional and structural factors that may need additional support."

Comment 28. Nonetheless, relying solely on primary care to bring a strong message to patients with MASLD and initiate a holistic treatment plan involving a tailored multidisciplinary team is not feasible due to time constraints and specialized aspects of follow-up.

- Great point.

Author's reply: Thank you.

Comment 29. In addition, we expect policymakers to actively promote disease destigmatization while limiting promotion and advertising of calorie-rich and highly processed foods.

- This conclusion statement seems awkwardly phrased. "Calorie rich" could also be "nutrient dense" and thus good. The connection between promotion and advertising is not otherwise

explored in the article and could raise issues (e.g., protected commercial speech, regulatory definitions of "healthy") not directly connected to liver health outcomes.

Not sure the final sentence is needed. The intent is correct but the language should be avoided or clarified.

Author's reply: Thank you for this comment. We have removed the final sentence, so the article can end on a positive note about multimodal treatment plans, which is the core of the manuscript: "In 5 years, we speculate that patient engagement in MASLD will be enabled in many countries through dedicated multimodal treatment plans."